# Treatments and Outcomes in Neuroendocrine Patients Treated with Long-Acting Somatostatin Analogues: An Italian Real-World Propensity Score-Matched Cohort Study

**DOI:** 10.3390/biomedicines13020515

**Published:** 2025-02-19

**Authors:** Nicoletta Ranallo, Andrea Roncadori, Nicola Gentili, William Balzi, Mattia Altini, Virginia Ghini, Roberta Maltoni, Alice Andalò, Martina Cavallucci, Maddalena Sansovini, Valentina Fausti, Maria Teresa Montella, Ilaria Massa, Valentina Danesi

**Affiliations:** 1Clinical and Experimental Oncology, Immunotherapy, Rare Cancers and Biological Resource Center, IRCCS Istituto Romagnolo per lo Studio dei Tumori (IRST) “Dino Amadori”, 47014 Meldola, Italy; nicoletta.ranallo@irst.emr.it (N.R.); virginia.ghini@irst.emr.it (V.G.); valentina.fausti@irst.emr.it (V.F.); 2Outcome Research, Healthcare Administration, IRCCS Istituto Romagnolo per lo Studio dei Tumori (IRST) “Dino Amadori”, 47014 Meldola, Italy; andrea.roncadori@irst.emr.it (A.R.); william.balzi@irst.emr.it (W.B.); roberta.maltoni@irst.emr.it (R.M.); direzione.sanitaria@irst.emr.it (M.T.M.); ilaria.massa@irst.emr.it (I.M.); valentina.danesi@irst.emr.it (V.D.); 3Data Unit, Healthcare Administration, IRCCS Istituto Romagnolo per lo Studio dei Tumori (IRST) “Dino Amadori”, 47014 Meldola, Italy; nicola.gentili@irst.emr.it (N.G.); martina.cavallucci@irst.emr.it (M.C.); 4Assistenza Ospedaliera Regione Emilia-Romagna, 40127 Bologna, Italy; mattia.altini@regione.emilia-romagna.it; 5Nuclear Medicine Unit, IRCCS Istituto Romagnolo per lo Studio dei Tumori (IRST) “Dino Amadori”, 47014 Meldola, Italy; maddalena.sansovini@irst.emr.it

**Keywords:** neuroendocrine tumour (NET), real-world evidence, somatostatin analogue (SSA), lanreotide, octreotide, propensity score matching (PSM), Italy

## Abstract

**Objectives**: The aim of this study was to investigate the treatment patterns and outcomes in two propensity score-matched cohorts of patients with neuroendocrine tumours (NETs) treated with first-line somatostatin analogue (SSA). **Methods**: Metastatic NET patients treated with first-line SSA (2009–2022) were retrospectively examined. First-line lanreotide vs. octreotide cohorts were matched 1:1 by propensity scores for demographics, tumour characteristics, and diagnosis year. Progression-free survival (PFS) and overall survival (OS) were analysed using Kaplan–Meier analysis and the Cox proportional hazards model. **Results**: Among 441 patients, 310 were matched (155 in both the octreotide and lanreotide groups). First-line SSA was monotherapy (63.5%) or combination with other medications (36.5%). A total of 77% of second-line patients (188/244) maintained their initial SSA medication in combination with other therapies. Radioligand therapy with lanreotide (N = 72; 29.5%) or octreotide (N = 70; 28.7%) was the most common second-line treatment. First-line lanreotide and octreotide cohorts had similar median PFS (15.5; 95% CI: 13.6–19.1 vs. 14.0; 95% CI: 12.0–15.8 months), despite octreotide having a 36% higher likelihood of moving to the second line than lanreotide (95% CI: 1.05–1.76, *p* = 0.018). Multiple metastases (HR = 1.45; *p* = 0.004, 95% CI: 1.13–1.87) and Ki-67 > 20% (HR = 2.34; *p* < 0.001, 95% CI: 1.43–3.83) were significantly associated with the worst PFS. First-line lanreotide patients had a median OS of 10.4 years (95% CI: 7.5-NA) and octreotide 9.2 years (95% CI: 7.3-NA) (*p* = 0.537). Bone metastases increased death risk by 91% (*p* = 0.014; 95% CI: 1.14–3.20). **Conclusions**: SSA monotherapy is the main first-line treatment and most subsequent treatments include SSA with additional medications. Cohorts had similar PFS/OS, but octreotide demonstrated a 36% significantly higher likelihood of moving to the second-line treatment.

## 1. Introduction

Neuroendocrine neoplasms (NETs) represent a rare and heterogeneous group of tumours that arise from neuroendocrine system cells involved in the release of peptides and amines with local activity [1].

In Italy, similar to numerous other countries, the incidence of NETs has significantly risen in recent decades, with a rate of 2–7 per 100.00 people [2,3,4,5,6,7]. The most common NETs arise from the gastroenteropancreatic (GEP) structures, accounting for 60% of cases, followed by the lung at approximately 30% [2,3,4]. About 12–20% of cases are metastatic at onset [8]. NETs are also classified as “functioning” or “non-functioning” according to their ability to secrete bioactive products that can lead to distinct clinical manifestations such as diarrhoea, flashes, and abdominal pain, known as carcinoid syndrome (CS) [9]. Neuroendocrine neoplasms (NENs) are classified based on histopathological and biological characteristics. The World Health Organization (WHO) GEP-NEN classification is based on the proliferative index (Ki-67 and mitotic count) and the degree of differentiation (well-differentiated tumours (G1 to G3) and poorly differentiated neuroendocrine carcinomas (NECs)) [10].

Somatostatin analogues (SSAs), such as octreotide and lanreotide, are used in NETs to reduce the overproduction of hormones and, consequently, their associated symptoms. SSAs can also slow tumour growth by directly binding somatostatin receptors, inhibiting angiogenesis, or inducing apoptosis [11].

The PROMID and CLARINET trials, using octreotide and lanreotide, respectively, demonstrated a significant efficacy of SSAs in progression-free survival (PFS) among patients with unresectable or metastatic well-differentiated NETs [12,13]. Treatment with SSAs is considered the standard first-line therapy for treating well-differentiated metastatic NETs due to their highly effective symptom and tumour growth control [14]. Although the role of SSAs is well recognised as the first-line treatment of NETs, the optimal treatment sequencing following the progression of SSAs remains unclear due to a lack of solid data and confusion in clinical practice guidelines [15]. The importance of real-world studies is particularly pronounced in uncommon diseases, as the execution of clinical trials may pose more substantial challenges [16]. The existing real-world studies on patients with NETs are mainly concentrated on claims database analysis [17,18,19,20,21,22]. However, the lack of clinical information, such as treatment duration and CS identification, may compromise the reliability of the claims reported in these investigations and limit the evaluation of PFS and OS. To overcome the constraints of existing research, it is essential to carry out real-world studies utilising more comprehensive clinical data, such as medical charts [23,24,25].

This study aimed to describe the real-world treatment patterns and clinical outcomes of metastatic NETs by comparing two propensity score-matched cohorts, lanreotide vs. octreotide, matched at the first-line treatment.

## 2. Materials and Methods

This analysis was a single-centre, retrospective observational study among NET patients conducted at IRCCS Istituto Romagnolo per lo studio dei Tumori (IRST) “Dino Amadori” of Meldola (Forlì-Cesena) to investigate the clinical outcomes and treatment pattern of NETs [26]. The eligible population and patient information were gathered retrospectively from Electronic Health Records (CCE Log80 2.6 of Log80 S.r.l) maintained during routine clinical practice. Initially, we identified patients with a structured diagnosis record annotated as “Apudoma”. Subsequently, a search using the regular expression (regex) technique was performed with Structured Query Language (SQL query, MySql 5.0.95) on clinical documents (clinical diaries, medical reports, and discharge letters) to identify potentially eligible patients by looking for text patterns. Specifically, the search included specific terms relative to somatostatin analogues and/or synonyms (lanreotide, octreotide, and their commercial names) in combination with other target strings such as NET and metastasis. Oncologists manually reviewed these patients to verify the eligibility criteria. Conversely, patients whose clinical records did not contain the above text patterns or contained the words paraganglioma and pheochromocytoma were excluded. Data collection was followed by data cleaning and supplemented by a manual review of unstructured data (i.e., clinical notes, radiology and pathological reports) to achieve a high completion rate for this analysis. The index date was defined as the initiation of a first-line SSA for the treatment of metastatic NETs between January 2009 and June 2022. Patients were followed from this date through various treatment lines until the earliest date, at the end of the observational period (June 2023), the last documented follow-up, or the patient’s death. Longitudinal data (i.e., both treatment and outcome data) were retrieved during the entire clinical pathway. More precisely, follow-up data were collected during outpatient visits, routine laboratory examinations, disease assessments, drug administration, and all the other procedures that a NET patient can receive in accordance with clinical guidelines and based on the patient’s specific needs. Outcomes were compared between patients who received first-line octreotide and those who received first-line lanreotide. This study was approved by the Scientific and Medical Committee and the Ethics Committee of IRST-IRCCS Area Vasta Romagna CEROM (approval number: 3219).

### 2.1. Study Population

The eligible population included the following:(a)Adult patients;(b)Patients with a metastatic diagnosis of gastroenteropancreatic neuroendocrine tumours (GEP-NETs) according to the 2017–2019 WHO classification [27,28] or pulmonary neuroendocrine tumours according to the 2015 WHO classification [29];(c)Patients who initiated treatment with long-acting release (LAR) lanreotide or octreotide (as monotherapy or in combination with other therapeutic drugs for at least three cycles) as a first-line agent or to manage the symptoms of CS between January 2009 and June 2022;(d)Patients with at least one medical visit at the IRST Institution during the accrual period.

Patients who initiated SSA medication before the metastatic diagnosis or after the administration of first-line treatment were excluded.

### 2.2. Statistical Analysis

Patients who underwent first-line treatment with SSA (octreotide or lanreotide) were matched 1:1 using a nearest neighbour propensity score with a maximum distance defined as a calliper of 0.2 times the pooled standard deviation of the logit of the propensity score. The propensity of initiating treatment with the molecule with the lowest frequency was estimated using a logistic regression model in which index year (i.e., the year of diagnosis), sex, age at index date, primary tumour localization, tumour grade (WHO 2017–2019 classification), Ki-67, metastasis localization, carcinoid syndrome, carcinoid heart, and surgical procedures were included as regressors. Population size did not depend on any statistical power computation, and all consecutive patients who met the inclusion and exclusion criteria during the accrual period were included. Patients were grouped into two cohorts, hereinafter octreotide and lanreotide cohorts, based on the treatment received on the index date (i.e., index treatment).

Means, standard deviations (SD)s, medians, and ranges were used to describe quantitative variables as appropriate. Absolute frequencies, together with proportions, were used to describe categorical variables. Patient demographics and clinical characteristics were represented at the time of NET metastatic diagnosis. A Sankey treatment sequence plot was drawn to show treatments by the line of therapy over time. Time-to-event variables (e.g., for determining overall survival (OS) and PFS) were summarised using Kaplan–Meier (KM) survival estimates; group comparisons were performed using log-rank tests. Furthermore, Cox proportional hazard (PH) regression models were developed to investigate the factors associated with time to next treatment (TTNT), PFS, and OS. All *p*-values less than 0.05 were considered statistically significant, and tests, unless otherwise specified, were two-tailed. Missing data were expected to be entirely random (MCAR). Missing data at the patient’s accrual were accounted for while performing the propensity score matching (PSM) (i.e., baseline missing data were used as informative data for propensity score estimation). Statistical analysis was performed using R statistical software (www.r-project.org) version 3.6.3.

## 3. Results

### 3.1. Patients Characteristics

According to the search query, 801 patients were identified as potentially eligible. After a manual review, 441 individuals were confirmed eligible. A flowchart illustrates a detailed breakdown of the case selection process (Figure 1).

Among eligible patients (N = 441), 262 (59.4%) received octreotide and 179 (40.6%) lanreotide treatment either alone or in combination with another drug as the initial treatment. After PSM application (see Appendix A for the multivariable logistic regression model details) and the common support requirement fulfilment assessment (Appendix A), the final cohort consisted of 310 patients: 155 treated with octreotide in the first line and the remaining 155 with lanreotide. Table 1 summarises the demographic and clinical characteristics of pre- and post-matched populations.

Specifically, the overall distance in terms of propensity scores among matched and unmatched individuals in the lanreotide and octreotide groups is depicted in the Appendix A. After the PMS, the *p*-values were not statistically significant (*p* > 0.05), indicating that the variables were evenly distributed between the two similar cohorts.

Focusing on the post-matching population, most patients in both cohorts were male (>58.1%). The mean age at diagnosis of metastatic disease was 60.0 years (SD = ±12.4) for patients treated with octreotide and 59.8 years (SD = ±11.6) for patients treated with lanreotide. In the population treated with octreotide, the primary tumour originated in the gastrointestinal tract (43.2%), followed by the pancreas (37.4%). Conversely, in the population treated with lanreotide, the most common primary site was the pancreas (43.2%), followed by the gastrointestinal tract (38.7%). In both cohorts, pathology showed that more than 60.0% of the patients had the Ki-67 proliferation index between 3% and 20%. The most common metastasis sites were the liver (>86.5%) and/or the lymph nodes (>49.7%). CS was present in about 37% of patients, while carcinoid heart was recorded in 1.9% and 3.2% of the octreotide and lanreotide cohorts, respectively. Overall, 27.7% of patients in the octreotide group and 15.2% in the lanreotide group underwent surgery before metastatic disease.

### 3.2. Pattern Treatment Sequence

Approximately 63.5% of patients initiated octreotide (N = 103) or lanreotide (N = 94) as monotherapy in the first-line setting (Figure 2). A minority of patients (N = 113; 36.5%) were treated with SSAs in combination with one or more agents, such as chemotherapy (singlet, doublet, or triplet), mammalian target of rapamycin (mTOR) inhibitors, tyrosine kinase inhibitors, or peptide receptor radionuclide therapy (PRRT) (Figure 2).

In total, 244 (244/310, 78.7%) patients received second-line therapy (Appendix A). Of these, 188 (188/244; 77.0%; 102 for lanreotide plus 86 for octreotide) maintained the SSA medication initiated in the first line (Appendix A). A switch in SSA from the first line to the second line was observed in 29 (29/244, 11.9%) patients (7 from lanreotide to octreotide; 22 vice versa) (Appendix A). Additionally, 27 patients (27/244, 11.1%) discontinued the SSA (12 and 15 for lanreotide and octreotide, respectively) (Appendix A). The most widely used medications in the second line were lanreotide (N = 72; 29.5%) and octreotide (N = 70; 28.7%) in combination with PRRT, both with and without chemotherapy (Appendix A). In subsequent lines, no particular treatment patterns were shown, given the small number of patients treated. In addition, the treatment patterns were reported according to NET sites, split by the gastrointestinal tract, lung, others/unknown, and pancreas (Appendix A).

### 3.3. Time to Next Treatment

As can be deduced from the KM plot, patients receiving first-line lanreotide had a median time of 20.9 (95% CI: 17.0–25.5) months to start second-line treatment, whereas those using octreotide had 16.9 months (95% CI: 14.9–20.4) (*p* = 0.055) (Figure 3a).

According to the multivariable Cox’s PH regression model, with other patient characteristics kept fixed, patients treated with octreotide had a significantly higher likelihood of moving to the second line of treatment, by 36% (hazard ratio (HR) 95% CI: 1.05–1.76; *p* = 0.018), than those who received first-line lanreotide medication (Figure 3b). Variables associated with the earlier initiation of second-line treatment included a high expression of Ki-67 > 20% (HR = 2.49; 95% CI: 1.48–4.21; *p* < 0.001) and the presence of liver (HR = 1.75; 95% CI: 1.17–2.62; *p* = 0.006) and lymph node (HR = 1.31; 95% CI: 1.01–1.71; *p* = 0.044) metastases. On the contrary, a low Ki-67 value < 3% (HR = 0.54; 95% CI: 0.40–0.74; *p* < 0.001) was associated with a reduced risk of starting a second line.

### 3.4. Progression-Free Survival

Patients who received lanreotide treatment had a median PFS of 15.5 (95% CI: 13.6–19.1) months, while those treated with octreotide had a median PFS of 14.0 months (95% CI: 12.0–15.8) (*p* = 0.074) (Figure 4a).

Patients treated with octreotide exhibited a 34% (HR 95% CI: 1.06–1.71; *p* = 0.016) increased risk of progression than those treated with lanreotide (Figure 4b). Variables that have been found to increase the likelihood of disease progression were the presence of multiple metastases (HR = 1.45; 95% CI: 1.13–1.97; *p* = 0.003) and a Ki-67 score higher than 20% (HR = 2.34; 95% CI: 1.43–3.82; *p* < 0.001).

### 3.5. Overall Survival

This study found that patients treated with first-line lanreotide had a median OS of 10.4 (95% CI: 7.5-NA) years, whereas the other group demonstrated a median of 9.2 (95% CI: 7.3-NA) years (*p* = 0.537) (Figure 5a).

The first quartile for the octreotide group ended at 4.8 years (95% CI: 3.9–6.8), while it was 5.6 (95% CI: 4.2–7.5%) years for the lanreotide group. Five-year survival was estimated as 78.6% (95% CI: 70.5–87.6%) for patients who received lanreotide and 69.1% (95% CI: 59.9–79.8%) for the octreotide group (Appendix A). Among the various NET sites, except for the pancreas, the OS corresponding to first-line lanreotide was higher than that associated with first-line octreotide, despite the absence of statistical significance (Appendix A). Irrespective of the first-line treatment administered, the 5-year OS associated with the localization of the primary was similar between gastrointestinal tract (69.1%; 95% CI: 65.3–84.4%) and pancreas patients (73.1%; CI: 63.5–84.3%) (Appendix A). The 5-year survival rate reached 66.5% (95% CI: 44.0–100.0) and 87.1% (73.5–100.0) for lung and other/unknown patients, respectively (Appendix A). However, we did not observe statistical differences.

A statistically significant risk factor was age (HR = 1.05; 95% CI: 1.02–1.07; *p* < 0.001); each additional year of age was associated with a 5% increase in the risk of mortality (Figure 5b). Additionally, the presence of bone metastases (HR = 1.91; 95% CI: 1.14–3.20; *p* = 0.014) was a significant risk factor, with a 91% increase in the hazard of dying. On the other hand, the factors that were statistically significantly associated with a reduced risk of mortality were Ki-67 < 3% (HR = 0.47; 95% CI: 0.25–0.87; *p* = 0.017) and prior surgical treatment (HR = 0.42; 95% CI: 0.23–0.77; *p* = 0.005; 58% lower probability). Even if statistical significance was not observed, patients with a positive 18F-fluorodeoxyglucose (18F-FDG) PET had a worse prognosis in terms of OS (HR = 1.56; 95% CI: 0.94–2.56; *p* = 0.082) than patients with a positive 68-gallium positron emission tomography (68Ga-PET) (HR = 0.20; 95% CI: 0.07–0.62; *p* = 0.005).

## 4. Discussion

This retrospective matched cohort study analysed the treatment patterns, sequences, TTNT, PFS, and OS in patients with GEP-NETs or lung neuroendocrine tumours treated with octreotide versus lanreotide in a real-world setting. In general, even at the pre-matching stage, the patients’ characteristics were balanced between the octreotide and lanreotide groups, with the only exception being the NET diagnosis year, probably due to the different availability of the two drugs on the market or otherwise attributable to different prescribing behaviour over time.

This study demonstrates that SSA as monotherapy is the primary treatment choice for most patients, and many of those remain on SSA mono- or combination therapy during subsequent lines of treatment. As observed by a previous study, our data suggest a clear preference for continuing the same SSA even in second-line treatment, usually in combination with other regimens, particularly PRRT [24]. As recommended by guidelines, the utilisation of mTOR inhibitors and tyrosine kinase inhibitors is limited in the initial lines of treatment, since they are typically chosen for the treatment of progressive NETS [14]. Our findings indicated that median PFS was similar between the octreotide (14.0 months, 95% CI: 12.0–15.8) and lanreotide (15.5 months, 95% CI: 13.6–19.1) groups. The median time to discontinue the first line of treatment with lanreotide (20.9 months, 95% CI: 17.0–25.5) and octreotide (16.9 months, 95% CI: 14.9–20.4) was in line with the 15.0 and 18.6 months reported in previous studies [18,23,30]. The median survival times of 10.4 years (95% CI: 7.5-NA) for the lanreotide group and 9.2 years (95% CI: 7.3-NA) for the octreotide group are generally longer than those reported in the literature. Our findings are about double the OS values reported in some studies [5,31] and still longer than previous investigations that estimated a survival rate between 6.7 and 8.2 years [21,32]. These differences in OS may be attributed to the combination of several factors. More precisely, this research study was conducted in a highly specialised centre with extensive expertise in NET management [33]. This setting contributed to including a patient cohort with access to advanced therapies, including PRRT. In contrast, the other studies included a broader population from national databases or registries with heterogeneous healthcare service access and treatment quality. Furthermore, these papers included patients treated in earlier eras with less advanced therapeutic options. Specifically, the data from Dasari et al. reflect treatment patterns from the 1980s to 2012 [5], whereas Lesén et al. focused on patients diagnosed between 2005 and 2013 [32]. Conversely, our study examined treatment patterns between 2009 and 2022, including the most advanced therapeutic options such as targeted therapies, PRRT, and innovative surgical approaches. Undoubtedly, these therapeutic advances may have caused an improvement in survival trends in recent years. Additionally, our study’s application of propensity score matching minimised confounding factors, enabling a more accurate estimation of OS compared to studies with heterogeneous cohorts. On the other hand, Kulke calculated a more prolonged survival of 14.9 years when both metastatic and advanced NETs were present [34].

Based on multivariable modelling, bone metastases are associated with an increased risk of death compared to other distant metastasis sites (HR 1.91; 95% CI: 1.38–3.20). This finding is in contrast to a prior investigation in which the presence of liver metastases was associated with higher mortality risk [32]. A more prolonged survival among surgically treated patients has also been documented previously [31]. The 5-year survival rate for the gastrointestinal tract is 74.2% (95% CI: 65.3–84.4%), which aligns with a previous study that reported a rate of approximately 70% [31]. Conversely, the literature indicates that pancreatic neuroendocrine tumours have the lowest 5-year survival rate, reported at less than 40%, significantly lower than our finding of 66.5% [31,35]. Nonetheless, this discrepancy may be attributed to the lack of updated studies in the literature and the observation of a continuous improvement in survival rates across the years in the SEER database [36]. As we know, well-differentiated NETs often express somatostatin receptors and, in most cases, demonstrate radiological positivity in PET investigations [37]. 68Ga-PET is one of the most used tests for the management of NET patients and is essential for selection for radioreceptor therapy [38]. 18F-FDG PET is a helpful practice in diagnosing and examining poorly differentiated neuroendocrine neoplasms. Well-differentiated forms have modest 18F-FDG PET positivity. Our study, as previously demonstrated, confirmed that the positivity of the tumour on 18F-FDG PET represents a negative prognostic factor and allows for the identification of poorly differentiated cell clones and, typically, is correlated with a higher Ki-67 [39]. The combined use of both functional imaging methods (68Ga and FDG PET/CT) allows NENs to be evaluated and classified optimally and, consequently, allows the practitioner to choose the best treatment for the patient.

The strength of this study is the use of a propensity score matching method, which minimises the discrepancies between the lanreotide and octreotide patient groups and enables balanced and unbiased comparison.

The study presents some limitations, particularly the retrospective nature and the cohort of patients limited to a single centre. However, since IRST is an expert member of the EURACAN consortium—the ERN (European Reference Network) for Rare Adult Solid Tumours—it is a highly attractive specialised centre for the Italian NET population [33]. Indeed, the study population was highly heterogeneous in geographical origin (covering the entire national territory), reflecting our centre’s reputation as a leading centre for NET care, especially gastrointestinal NET. Indeed, many patients are referred to IRST for PRRT, a standard treatment predominantly used for GEP-NETs. This referral pathway likely accounts for the under-representation of lung NETs, which led to an overestimation of OS in our study cohort.

A significant number of patients presented incomplete treatment information because they were lost to follow-up, which may have led to our data misrepresenting the actual SSA exposure. Similarly, the retrieval of data on comorbidities and performance status could have reduced the results’ uncertainty.

The extreme heterogeneity of neuroendocrine tumours, the prolonged survival of patients, and the diversity of treatments and combinations represent further difficulties in retrospective studies of these tumours. However, the high number of patients enrolled represents a good study sample for these uncommon tumours. Utilising a robust propensity score-matched cohort, we compared the effectiveness of octreotide and lanreotide in a diverse patient population, revealing that different treatments result in comparable progression-free and overall survival rates, even if octreotide exhibited a 36% significantly greater likelihood of transitioning to second-line treatment compared to lanreotide. Furthermore, our study reveals critical factors influencing clinical outcomes, such as the effects of numerous metastases and elevated Ki-67 indices on progression-free survival, whereas bone metastases increased mortality risk by 91%. These data certainly need further confirmation from multicenter and prospective studies.

## Figures and Tables

**Figure 1 biomedicines-13-00515-f001:**
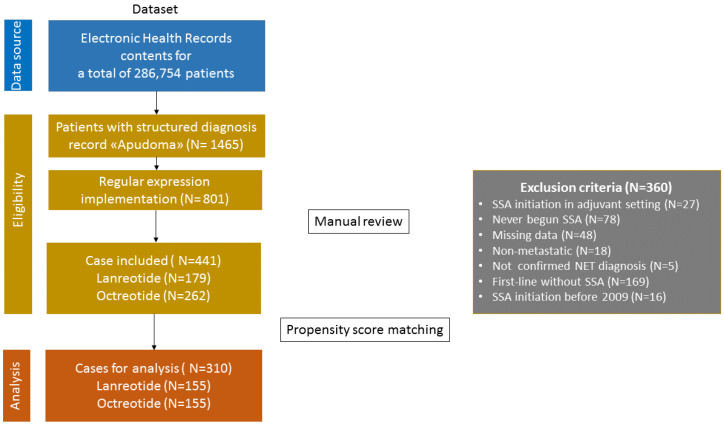
The flowchart of the process of case selection.

**Figure 2 biomedicines-13-00515-f002:**
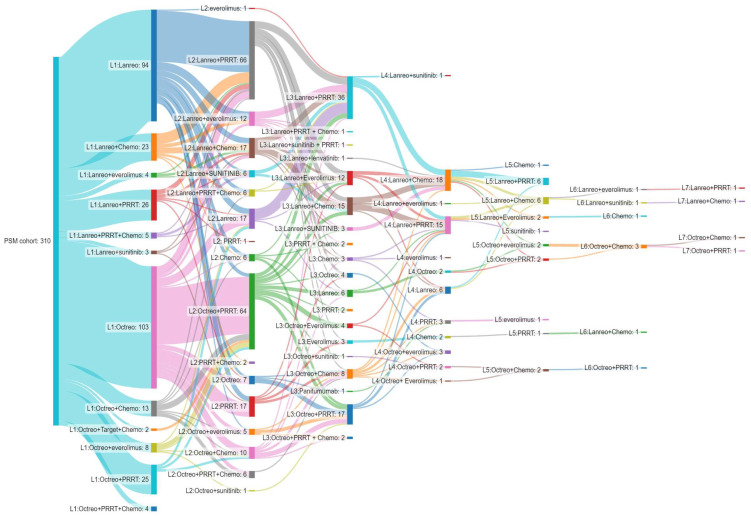
Sankey diagram showing the treatment patterns and sequences of the post-matching cohort (N = 310) from the first line (L1) up to the seventh line (L7) of treatment. Chemotherapy treatment (Chemo) includes the following agents: capecitabine, oxaliplatin, etoposide, temozolomide, carboplatin, streptozotocin, fluorouracil, doxorubicin, dacarbazine, epirubicin, and irinotecan. Abbreviations: PSM, propensity score matching; Octreo, octreotide; Lanreo, lanreotide; Chemo, chemotherapy; PRRT, peptide receptor radionuclide therapy.

**Figure 3 biomedicines-13-00515-f003:**
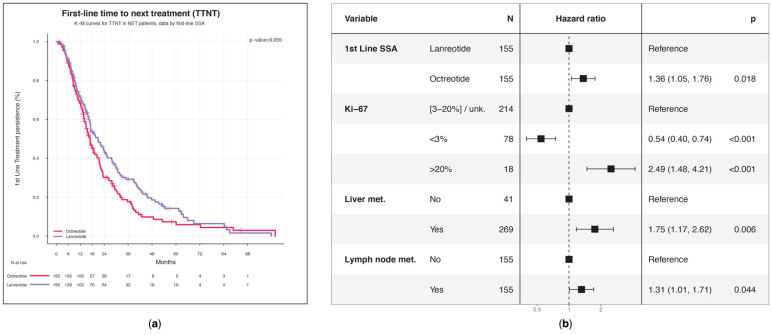
First line time to next treatment (TTNT): (**a**) KM estimates of first-line treatment persistence. KM curves are drawn separately for first-line SSA treatment. Censored observations are represented with a vertical line in the curves at the time of censoring; (**b**) Multivariable Cox PH regression model for estimating the effect of baseline characteristics on first-line time to subsequent treatment.

**Figure 4 biomedicines-13-00515-f004:**
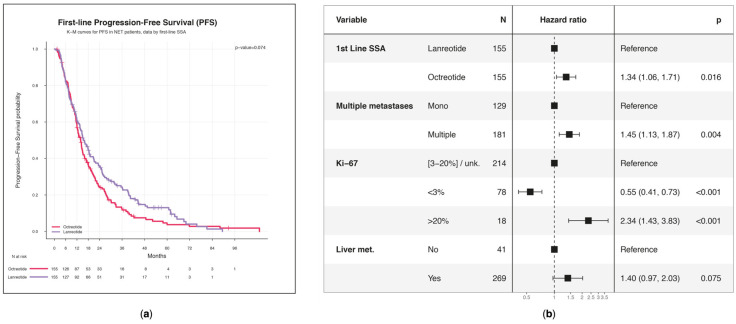
Progression-free survival (PFS): (**a**) KM estimates of PFS curves associated with first line grouped by SSA treatment. Censored observations are represented with a vertical line in the curves at the time of censoring; (**b**) multivariable Cox PH regression model for estimating the effect of baseline characteristics on PFS.

**Figure 5 biomedicines-13-00515-f005:**
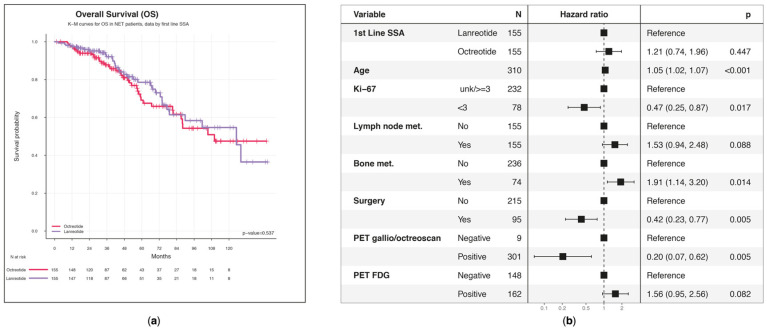
Overall survival (OS): (**a**) KM estimates of OS according to first-line SSA treatment. Censored observations are represented with a vertical line in the curves at the time of censoring; (**b**) multivariable Cox PH regression model for estimating the effect of baseline characteristics on OS.

**Table 1 biomedicines-13-00515-t001:** Baseline demographics and clinical characteristics of patients pre- and post-matching.

	Pre-Matching			Post-Matching		
Characteristics	1L Octreotide N = 262 (%)	1L Lanreotide N = 179 (%)	*p*-Value	1L Octreotide N = 155 (%)	1L Lanreotide N = 155 (%)	*p*-Value
Sex			0.4915			0.7290
Females	107 (40.8)	79 (44.1)		62 (40.0)	65 (41.9)	
Males	155 (59.2)	100 (55.9)		93 (60.0)	90 (58.1)	
Age: mean ± SD	60.9 ± 11.8	59.1 ± 11.9	0.1151	60.0 ± 12.4	59.8 ± 11.6	0.9095
MEN-1 syndrome	9 (3.4)	2 (1.1)	0.1254	3 (1.9)	2 (1.3)	0.6521
NET site			0.1005			0.8915
Lung	22 (8.4)	16 (8.9)		13 (8.4)	12 (7.7)	
Pancreas	81 (30.9)	76 (42.5)		58 (37.4)	67 (43.2)	
Gastrointestinal tract	126 (48.1)	68 (38.0)		67 (43.2)	60 (38.7)	
Others	5 (1.9)	5 (2.8)		5 (3.2)	5 (3.2)	
Unknown origin	28 (10.7)	14(7.8)		12 (7.7)	11 (7.1)	
Ki-67 proliferation index						
<3%	68 (26.0)	47 (26.3)	0.7477	39 (25.2)	40 (25.8)	0.9217
3–20%	161 (61.5)	104 (58.1)		98 (63.2)	94 (60.6)	
>20%	14 (5.3)	12 (6.7)		8 (5.2)	9 (5.8)	
Unknown	19 (7.3)	16 (8.9)		10 (6.5)	12 (7.7)	
Metastasis sites						
Liver	224 (85.5)	157 (87.7)	0.5056	134 (86.5)	135 (87.1)	0.8669
Lung	10 (3.8)	10 (5.6)	0.3804	7 (4.5)	7 (4.5)	1.0000
Lymph nodes	128 (48.9)	91 (50.8)	0.6825	78 (50.3)	77 (49.7)	0.9096
Bones	55 (21.0)	45 (25.1)	0.3071	37 (23.9)	37 (23.9)	1.0000
Peritoneum	37 (14.1)	25 (14.0)	0.9632	20 (12.9)	20 (12.9)	1.0000
Other	27 (10.3)	27 (15.1)	0.1328	17 (11.0)	17 (11.0)	1.0000
Carcinoid syndrome	92 (35.1)	68 (38.0)	0.5376	58 (37.4)	59 (38.1)	0.9067
Carcinoid heart	8 (3.1)	6 (3.4)	0.8606	3 (1.9)	5 (3.2)	0.4737
Surgery prior metastatic diagnosis	90 (34.4)	52 (29.1)	0.2420	43 (27.7)	42 (15.2)	0.2675
Year of NET diagnosis			0.0002			0.3843
1996–2010	46 (17.6)	16 (8.9)		17 (11.0)	16 (10.3)	
2011–2013	87 (33.2)	38 (21.2)		37 (23.9)	38 (24.5)	
2014–2016	89 (34.0)	53 (29.6)		63 (40.6)	49 (31.6)	
2017–2019	32 (12.2)	56 (31.4)		30 (19.3)	41 (26.5)	
2020–2022	8 (3.0)	16 (8.9)		8 (5.2)	11 (7.1)	

## Data Availability

The raw data presented in this study are available at a reasonable request from the corresponding author.

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
