# Peer review of "Treatments and Outcomes in Neuroendocrine Patients Treated with Long-Acting Somatostatin Analogues: An Italian Real-World Propensity Score-Matched Cohort Study"

_biomedicines, 2025, doi:10.3390/biomedicines13020515_

Round 1
Reviewer 1 Report
Comments and Suggestions for Authors
the authors present an analysis of neuroendocrine patients treated with SSA's using a propensity score matching to establish the study cohort
for additional discussion:
1. while the authors state that lung is typically 30% of NET's, it is only <8% in this study...comments??? and impact for generalization??
2. the pre-match vs post-match comparison shows that GI tract NET's are "overtreated" with Octreotide...and this is adjusted for in developing the cohort...but what does this reflect about the guidelines and current standard of care?
Author Response
COMMENT 1: while the authors state that lung is typically 30% of NET's, it is only <8% in this study...comments??? and impact for generalization??
AUTHORS’ REPLY: We thank the reviewer for raising this important point. The IRST is an expert member of the EURACAN consortium, part of the European Reference Network (ERN) for Rare Adult Solid Tumors (https://www.euracan.eu/find-an-expert-centre), and serves as a highly specialized referral center for the Italian NET population. Additionally, many patients are referred to IRST specifically for Peptide Receptor Radionuclide Therapy (PRRT), a standard treatment predominantly used for gastroenteropancreatic NET. Consequently, an underrepresentation of lung NETs in our study population may have occurred. Furthermore, patients undergoing PRRT generally come to IRST after a first opinion visit in other centres that suggest our Institute’s care to patients who will likely be eligible for this treatment. For what concerns the effect of lung NET underrepresentation, the literature (e.g., Dasari A. et al., Trends in the Incidence, Prevalence, and Survival Outcomes in Patients With Neuroendocrine Tumors in the United States, Jama Oncology, 2017) demonstrates that lung NETs have a worse overall prognosis than gastrointestinal NETs. This may have caused an overestimation of our study cohort's overall survival (OS). The authors added a comment from lines 356 to 364 in the discussion section to further explain this and highlight the potential impact of generalization.
COMMENT 2: the pre-match vs post-match comparison shows that GI tract NET's are "overtreated" with Octreotide...and this is adjusted for in developing the cohort...but what does this reflect about the guidelines and current standard of care?
AUTHORS’ REPLY: The authors appreciate the reviewer's comment. The observed pre-match overrepresentation of gastrointestinal-tract NETs treated with octreotide does not reflect any intentional preference by prescribing physicians. Many patients referred to our center from external institutions often arrived with first-line therapy already initiated, and the reasons behind the selection of octreotide in these cases are unknown. This underscores the retrospective nature of our study and the inherent variability in real-world treatment patterns, which our matching process has appropriately accounted for to ensure balanced comparisons.

Reviewer 2 Report
Comments and Suggestions for Authors
This study examined treatment patterns and outcomes in patients with metastatic neuroendocrine tumors treated with either lanreotide or octreotide as first-line somatostatin analogue therapy. A total of 441 patients were analyzed, with 310 matched based on demographics and tumor characteristics. The study assessed progression-free survival and overall survival, revealing that both SSAs are effective in NET treatment. However, octreotide users transitioned to second-line therapy more frequently. Due to the retrospective nature and missing data, a prospective, multicenter study is necessary to confirm these findings and investigate the underlying reasons for treatment shifts.
This is a well-conducted study with valuable clinical insights, making it suitable for publication. However, some minor but important issues should be addressed:
The study does not fully account for variability in tumor behavior, prior treatments, and patient conditions, which could influence treatment outcomes.
The reported overall survival exceeds that of previous studies, but the authors do not provide a clear rationale beyond the general trend of improving survival rates. A comparison with other cohorts or an exploration of potential biases (patient selection, advances in supportive care) would strengthen the findings.
The reliance on historical data introduces risks of missing, incomplete, or inconsistent information, potentially impacting the validity of results. A discussion on data accuracy and how missing data were handled would enhance transparency.
While octreotide patients transitioned to second-line treatment more frequently, the study does not clarify whether this was due to lower efficacy, differences in physician preference, or patient-specific factors. Further explanation or sensitivity analysis would provide a more nuanced interpretation.
Addressing these points would strengthen the study’s conclusions and improve its overall impact.
Author Response
We thank the reviewer for their detailed and thoughtful feedback on our study. Below, we respond to each of the concerns raise.
COMMENT 1: Due to the retrospective nature and missing data, a prospective, multicenter study is necessary to confirm these findings and investigate the underlying reasons for treatment shifts
AUTHORS’ REPLY: We thank the reviewer for this comment. We know the potential issues connected with a single-centre study (increased risk of selection bias, reduction of external validity, and generalizability issues). For this reason, we reported it in the study limitations (please see lines 355-356 in the Discussion section). However, IRST is an expert member of the EURACAN consortium - the ERN (European Reference Network) for Rare Adult Solid Tumours (https://www.euracan.eu/find-an-expert-centre) and a highly attractive specialised centre for the NET Italian population. For this reason, the study population was highly heterogeneous in geographical origin (covering the entire national territory), reflecting its reputation as a leading center for NET care. (Please see the amended manuscript version in rows 356 to 361). Finally, concerning the missing data and its potential for biasing the results, the oncologists manually retrieved the EHR data (Electronic Health Records) to reduce this issue. Furthermore, patients with unknown key information (i.e. inclusion/exclusion criteria variables) were excluded from this study (see Figure 1 - patients selection process flowchart).
COMMENT 2: The study does not fully account for variability in tumor behavior, prior treatments, and patient conditions, which could influence treatment outcomes.
AUTHORS’ REPLY: We thank the reviewer for their insightful comments. In retrospective and real-world studies, retrieving comprehensive information on all factors that could influence patient outcomes is often challenging. However, we addressed this by collecting (both structured and unstructured) data and incorporating these variables into the propensity score matching process to balance the cohorts and account for variability in patient conditions in the developed models. Regarding prior treatments, it is important to note that current guidelines do not recommend adjuvant or neoadjuvant therapies for NET patients. As such, prior treatments are not a confounding factor in this context. However, we collected data on prior surgical treatments performed before the metastatic diagnosis, which were included in our analysis to account for their potential impact on patient outcomes.
Regarding tumour behaviour, although the absence of tumour grade limits the full assessment of tumour variability, including the ki-67 proliferation index provides valuable insights into tumour behaviour. This key prognostic biomarker is widely used to analyse tumour aggressiveness and inform clinical decision-making in oncology. Lastly, we collected the MEN-1 syndrome, the primary tumour and the metastasis sites. Unfortunately, patients’ information on comorbidities or performance status was not available. We added this limitation to the discussion section (please see lines 366-368).
COMMENT 3: The reported overall survival exceeds that of previous studies, but the authors do not provide a clear rationale beyond the general trend of improving survival rates. A comparison with other cohorts or an exploration of potential biases (patient selection, advances in supportive care) would strengthen the findings.
AUTHORS’ REPLY: We appreciate the reviewer’s thoughtful comment and the opportunity to discuss the observed differences in Overall Survival (OS) further. Specifically, the longer OS observed in our study compared to previously published studies is an important finding, likely attributed to differences in the patient population, advancements in NET management over time, and methodological approaches.
Our study was conducted in a highly specialized center with extensive neuroendocrine tumour (NET) management expertise. This setting contributed to including a patient cohort with access to advanced therapies, such as peptide receptor radionuclide therapy (PRRT), which is unavailable in many centers. In contrast, the references cited in our manuscript included a broader population retrieved from the SEER, the French National Claims Database or the Swedish registries with heterogeneous healthcare access and treatment quality. Moreover, the cited studies included patients treated in earlier eras with less advanced therapeutic options and care pathways than our study cohort. For example:
- Dasari et al. analyzed SEER data reflecting treatment patterns from the 1980s to 2012.
- Lesén et al. focused on patients diagnosed between 2005 and 2013
Conversely, our study analysed patients treated between 2009 and 2022, reflecting the more recent advancements in treatment (including targeted therapies, PRRT, and better surgical interventions).
Neither study employed propensity score matching, which may have led to imbalances in patient characteristics and treatment variables.
In conclusion, the higher OS observed is likely attributable to differences in the patient population, advancements in NET management over time and methodological strengths, including propensity score matching.
We added these considerations in the Discussion section to help readers better understand the context of our research and the precautions needed when comparing it with other studies (please see lines 305 to 319).
COMMENT 4: The reliance on historical data introduces risks of missing, incomplete, or inconsistent information, potentially impacting the validity of results. A discussion on data accuracy and how missing data were handled would enhance transparency.
AUTHORS’ REPLY: We thank the reviewer for this comment, which helped us to further reflect on the validity of the results.
We took specific measures to enhance data accuracy and transparency during our study design and data collection phases. Specifically, when data from structured variable fields were missing, oncologists manually retrieved information from Electronic Health Records (EHR) to reduce missingness while ensuring the completeness of key clinical information. Additionally, patients with insufficient or unknown essential variables — particularly those relevant to the inclusion and exclusion criteria — were excluded from the study to maintain the integrity of the dataset (see Figure 1 for the patient selection process flowchart). Moreover, as detailed in the “Statistical method section”, missingness in other than I/E criteria data was considered to be completely at random (MCAR) - see “Statistical method” section from line 157 to line 160. Finally, the missingness of data on variables used for estimating the propensity scores was used by categorizing variables (e.g., Ki-67%: <3%, 3-20%, >20%, unk.). Therefore, we argue that all these steps aimed to minimize the impact of missing data on our findings.
COMMENT 5: While octreotide patients transitioned to second-line treatment more frequently, the study does not clarify whether this was due to lower efficacy, differences in physician preference, or patient-specific factors. Further explanation or sensitivity analysis would provide a more nuanced interpretation.
AUTHORS’ REPLY: We thank the reviewer for the comment. Determining why patients treated with octreotide in the first-line are more likely to move to the second-line than lanreotide is complex. Switching to the second line is a multifactorial phenomenon, and several factors, including disease progression, treatment-related toxicities, clinical considerations, and preference choices, influence it.
In our study, 36.5% of patients received first-line therapy in combination with other treatments, which makes it challenging to isolate the specific effect of somatostatin analogues (SSAs) from the other agents. Consequently, this limitation complicates evaluating whether the differences in second-line therapy transitions were driven by SSA efficacy issues.
Furthermore, as mentioned above, IRST is a referral center of excellence for NETs; many of our patients were referred from external centers where the first-line treatment decisions had already been made. As a result, the rationale behind the initial choice of somatostatin analogue (octreotide vs. lanreotide) and the potential physicians' and/or patients' preferences may be unknown.
Nevertheless, the best response data during the first line were available. Certainly, the authors are aware that the best response during the first line does not fully respond to the reviewer’s comment. However, we found it interesting to analyze this data, which can give us useful information and some indications. While this information was not included in the results section due to both the above mentioned reason (i.e., the best response does not take into account patients/oncologist choices) and space limitations, we provide it here:
|
Best Response |
Octreotide (N) |
Octreotide (%) |
Lanreotide (N) |
Lanreotide (%) |
|---|---|---|---|---|
|
NA |
4 |
2,58% |
5 |
3,23% |
|
PD |
72 |
46,45% |
57 |
36,77% |
|
PR |
11 |
7,10% |
18 |
11,61% |
|
SD |
65 |
41,94% |
68 |
43,87% |
|
UNK |
3 |
1,94% |
7 |
4,52% |
The table above shows that a higher proportion of octreotide-treated patients (46.45%) had progressive disease (PD) than lanreotide-treated patients (36.77%), which may partly explain the higher rate of transition to second-line therapy in this group. However, both groups showed high (and similar) rates of Stable Disease (SD) (41.94% for octreotide and 43.87% for lanreotide), reflecting the effectiveness of both agents in disease control.

Reviewer 3 Report
Comments and Suggestions for Authors
Overall the study is interesting but still some limitations that I hope can be removed during the revision stage.
- Why is the study single centered? Would it be better to make it multicenter to achieve consensus results?
- Please mention the review committee number of this manuscript.
- Why is only adult population targeted?
- The figures are not upto standards. They can be improved for resolution.
- NET?
- The eligbility criteria not specified properly.
- The follow up protocol is not fully explained.
Author Response
COMMENT 1: Why is the study single centered? Would it be better to make it multicenter to achieve consensus results?
AUTHORS’ REPLY: We thank the reviewer for this comment. We know the potential issues connected with a single-centre study (increased risk of selection bias, reduction of external validity, and generalizability issues). For this reason, we reported it in the study limitations (please see lines 347-348 in the Discussion section). However, IRST is an expert member of the EURACAN consortium - the ERN (European Reference Network) for Rare Adult Solid Tumours (https://www.euracan.eu/find-an-expert-centre) and a highly attractive specialised centre for the NET Italian population. For this reason, the study population was highly heterogeneous in geographical origin (covering the entire national territory), reflecting its reputation as a leading center for NET care. (Please see the amended manuscript in rows of 348 to 353 lines).
COMMENT 2: Please mention the review committee number of this manuscript.
AUTHORS’ REPLY: We thank the reviewer for the comment. We added the Ethical Committee approval number in the Materials and Methods section. Please see line 110.
COMMENT 3: Why is only adult population targeted?
AUTHORS REPLY: We appreciate the reviewer’s valuable observation. The study is focused exclusively on adult patients because our institution is a scientific research center specializing in oncology treatment, research, and training, specifically for adult populations. The center does not treat minor patients.
COMMENT 4: The figures are not upto standards. They can be improved for resolution.
AUTHORS’ REPLY: We appreciate the reviewer’s thoughtful comment. As suggested, we improved the resolution of the figures 2, 3, 4 and 5.
COMMENT 5: NET?
AUTHORS’ REPLY: As suggested by the reviewer, the NET was added in the abbreviation section of the manuscript.
COMMENT 6: The eligbility criteria not specified properly.
AUTHORS’ REPLY: As suggested by the reviewer, we rewrote the eligibility criteria according to the study protocol section. Furthermore, Figure 1 - patients' selection process flowchart reported additional information on patient exclusion criteria.
COMMENT 7: The follow up protocol is not fully explained.
AUTHORS REPLY: There was no study-specific follow-up protocol. Indeed, the study's observational nature reflects the clinical routine practice following the guidelines and based on the specific patient’s needs. Furthermore, we included a sentence in the material and methods section to better clarify this point (Please see lines 102 to 107).
